# The Regionalization Process for Universal Health Coverage in Brazil (2008–2015)

**DOI:** 10.3390/healthcare9101380

**Published:** 2021-10-15

**Authors:** Rafael da Silva Barbosa, Eduardo Fagnani

**Affiliations:** 1Social Policy Graduate Program, Department of Social Services, Goiabeiras, Federal University of Espírito Santo, Vitória 29075-910, Brazil; 2Economic Development Graduate Program, Institute of Economics, Barão Geraldo, State University of Campinas, Campinas 13083-970, Brazil; eduardo.fagnani@uol.com.br

**Keywords:** health policy, universal health coverage, unified health system, installed capacity, cluster

## Abstract

The objective of this article is to analyze the development of the public and private offer for the universalization of health services, specifically, for the progression of the public network. The time period examined is from 2008 to 2015, when there was considerable economic growth and expansion of private health insurance and an unprecedented historical period with economic growth and reduction of social inequality. Across 5570 municipalities, the multivariate analysis model was used to estimate the level of concentration and the imbalance (heterogeneity) of installed health capacity of the network of health care services. Public spending on investment and human resources showed positive variation in all regions and in almost all population strata. The offer by the Unified Health System (public) of primary health care increased by 8000 new establishments in all regions, especially in previously uncovered cities and cities that had shortages of public health services. Public universalization almost reached its maximum, with about 70% of municipalities. The only setback was the significant reduction of 50% in the number of private establishments in primary health care services. The data suggest a positive movement toward the universalization of health services in Brazil, with the concentration of high-complexity care and the heterogeneity of the installed capacity being points for improvement.

## 1. Introduction

In Brazil, the Constitution establishes in its Article 196 that “Health is a right of all and a duty of the State and shall be guaranteed by means of social and economic policies aimed at reducing the risk of illness and other hazards and at the universal and equal access to actions and services for its promotion, protection and recovery” [1]. The Federal Constitution of Brazil was promulgated on 5 October 1988, and it is relatively new in comparison with other countries, with its few years of existence, as is the regionalization of health in Brazil. Since the redemocratization of the country, which culminated in the creation of the Unified Health System (SUS) in 1988, there have been four main cycles [2], namely, absolute decentralization, relative decentralization, regional decentralization, and hierarchical regionalization.

From its inception, SUS regional policy had to overcome the contradiction between the centralization and decentralization of services. The expansion had to be guided by the decentralization of primary services not yet available in several regions of the country and, at the same time, centralize highly technological services without penalizing the most distant cities. This would ensure both the scale necessary for cost reduction and simultaneously increase full coverage of low-, medium-, and high-complexity care.

The economic and political context at the time opened possible areas for advancement [2]. The first cycle saw the simple transfer of competence from the federally centralized health services to subnational entities (states and municipalities). This absolute decentralization of health services was not accompanied by greater autonomy of fiscal revenues for states and municipalities; hence, the second cycle sought to formulate new mechanisms for tax and sector transfers to correct these disparities. The objective was to distribute resources according to the volume of installed capacity of each region. However, instead of correcting matters, this mechanism only reinforced the disparities with a new type of federal transfer reconversion. The rule ended up benefiting mainly the major cities holding the largest health care budgets.

This regional policy only achieved concrete results with the beginning of the third cycle, which forged the first effective mechanism for regional health care decentralization. The Basic Operational Norm, in 1996 [3], innovated by creating the fixed per capita transfer of the primary care level (in Portuguese, the Piso de Atenção Básica Fixo or, PAB-FIXO). The proportional correction introduced by this new mechanism permitted major adjustments to the regional differences. Unlike the previous cycle, the PAB-FIXO linked the federal transfer of resources to population size and not to the volume of the installed capacity and; in this way, it produced the desired effect.

The redistributive process created by the PAB-FIXO strengthened municipal actions in the less favored regions, which, when added to the good results of the Family Health Strategy [4], generated a change in the level of quality health care and coverage, proving itself to be the first qualitative instrument of a regional character for health [5].

In the wake of these innovations, the 2006 Pact for Health [6] modernized the concept of regional health care decentralization and began the fourth cycle. This perspective came to be understood as an association between technical capacity (economies of scale) and policy (federal pacts) to guarantee the investments and resources necessary for change [2] within a framework in which the federal entities were prohibited from taking on any responsibilities.

It was intended that, in this process, national and state strategies should prioritize investment projects that strengthened SUS regionalization. The integrated development plans (IDPs) would serve to operationalize actions, guaranteeing the mapping of the distribution and supply of health services in each regional area within the parameters of (a) the incorporation of technologies, permitting economies of scale, and (b) scope, ensuring equity of access [6].

The IDPs, together with the stabilization of the flow of resources secured by Constitutional Amendment 29, 2000 [7], consolidated the implementation of the fourth cycle. These factors ensured a broader and more integrated perspective for transformations to reduce regional inequalities, which are to be supported by the expansion of state and federal public investment [2] (p. 49).

In order to overcome fragmented health actions, the directives of Ordinance N°. 399 of the Pact for Health limited the operation of regional policy to parameters of scale and the federal pacts, in order to generate a hierarchy of actions [8]. The objective was to draw attention to the fact that the universalization of medium- and high-complexity care could only be achieved with the greater participation of the states and the Union.

The growth of the Brazilian economy between 2006 and 2014 allowed the implementation of the regional strategy of SUS public health policy. Economic booms contributed to increasing the effectiveness of the policy by alleviating the financial bottlenecks experienced by municipalities, states, and the Union [9].

Over the years, the results of regional health policy have been instrumental in reducing imbalances in the offer and in improving health conditions across all socioeconomic groups. According to a World Bank publication [4], in the 20 years of the building of the health system in Brazil, the SUS has brought regional gains of the highest relevance to the quality of health in Brazil. Geographic inequalities in health have been significantly reduced, with Northeast states receiving most of the benefits. Historically, the poorest regions of Brazil are the Northeast and North regions. Regional health policy has consolidated the universalization of primary health care and considerably expanded the medium-complexity outpatient network, reducing, in part, the pressure experienced by the municipalities in this specialism. Therefore, the analysis was made over an unprecedented historical period with economic growth and reduction of social inequality (Figure 1).

## 2. Materials and Methods

Studies on the Unified Health System (SUS) do not include their complementary elements within the private sector. These quantitative analyses evaluate SUS across its timeline without considering private participation. Although these studies characterize the public–private mixture profile of each segment of the Brazilian health system, estimates about the results are made separately [10]. We seek, therefore, to analyze the two partners (public and private) jointly across the 5570 Brazilian municipalities to understand the development of regional health policy.

Databases were extracted from the databases of the Brazilian Institute of Geography and Statistics (IBGE), Ministry of Health (MS), National Occupation Classification (CBO), and the National Agency for Supplementary Health (ANS) for the period from 2008 to 2015. Outlier methodology was used to calculate the quartiles (Q1 and Q3) and the interquartile interval (IQR) for a parameter of 1.5 to health expenditure. As these data are declarative, it is necessary to be more cautious with basic data. The law establishes that the respective federal transfers of health would only occur through the mandatory completion of health data by states and municipalities. Thus, each federal entity declares the information required by law. Despite the declaratory character of the information, this aspect qualitatively altered the robustness of the SIOPS database, which continues to guarantee the completion and continuity of the data series.

Of all variables analyzed (Table 1), expenditures on human resources, investment, primary, medium-, and high-complexity care are the most recent but least studied and published, within the scope of microdata for the 5570 municipalities. This is because it was only with the passing of Supplementary Law No. 141, dated 13 January 2012 [11], that it became mandatory to return these financial data as a condition for the transfer of funds. These data contribute to the improvement of information around the decentralization of SUS [12]. The data quality of the Public Health Budgeting System (SIOPS) has improved between 2008 and 2015. The payout was selected similar to the stage of budgetary (There are three stages of budget in Brazil: committed (empenhado), pay-off (liquidado) and pay-out (pago).) execution to analyze the data. For data treatment, the outlier methodology was used to smooth out possible distortions, since the source of information is declaratory, i.e., the municipalities are responsible for providing the data, not the federal government. Thus, there is no business intelligence (BI) for the process of collecting, organizing, analyzing, sharing, and monitoring information of SIOPS data.

For the quartiles (Q1 and Q3) and the interquartile interval (IQR) for a parameter of 1.5, where
IQR = Q3 − Q1;Lower outliers < Q1 − IQR × 1.5;Superior outliers > Q3 + IQR × 1.5,
observations outside the range [Q1 − IQR × 1.5 < data < Q3 + IQR × 1.5] were considered as outliers.

All the variables of the SIOPS database showed decreased amounts of absent (unavailable) information and presented better consistency within the information. Overall the number of “absent” and outliers fell by 6% (339–320) in human resource expenditure and 14% (615–528) in investment. The same happened with primary care and medium- and high-complexity care, which fell 20% (644–514) and 49% (3541–1792), respectively.

The regions that most improved the recorded data were the North and Northeast, in cities with less than 50,000 inhabitants.

The analysis of the data employed multivariate analysis to evaluate the specificities of the information within the regional context of the country. This approach was chosen because of its analytical capabilities to include “an ever-expanding set of techniques for data analysis that encompasses a wide range of possible research situations” [13] (p. 15). In this article, the techniques of factor and principal component analysis, Ward’s cluster analysis, and k-means clustering were applied to the Brazilian experience [14]. The technique is suggested due to its sharper focus. In view of many variables, it is possible to select the main variables relevant to the construction of the groups, moving toward the study’s goals.

Schematically, the objective of the factor and principal component analysis is “to find a way of condensing the information contained in a number of original variables into a smaller set of variables with a minimal loss of information” [13] (p. 16). When selecting the main variables, the technique for building the clusters was applied. Ward’s cluster analysis method, because of its hierarchical technique, assisted in the choice of the number of clusters to be formed. K-means clustering was then used for the definitive structuring of the clusters. In other words, the first step was applied principal component analysis to find the original variables; next and in the second step, standard Z was applied to the original variables to use Ward’s cluster analysis; third, Ward’s cluster analysis (hierarchical) was applied to build a dendrogram and visualize the best cut; finally, with the fourth step, the k-means clustering method was used for final clustering.

Table 2 shows the set of variables used in the factor analysis. The variables express demand (population) in relation to the supply factors of public installed capacity in health.

Factor analysis resulted in 3755 observations. According to Table 3, Factor 1 accounted for 74% of the total variability, while Factor 2 accounted for 14%, making clear the degree of importance (weight) of the first factor, in terms of the data variance, in relation to the second.

Table 4 provides the factorial loads for the selection of the original variables. The selection procedure was by the interpretation of the measurement of sampling adequacy (MSA), in which 0.80 or above is meritorious, 0.70 or above is middling, 0.60 or above is mediocre, 0.50 or above is poor, below 0.50 is unacceptable [13].

Thus, of the 15 variables listed for the grouping, 9 variables were most relevant to the formation of clusters.

The results of Table 4 indicate that the main relevant variables are population, human resource expenditure (SUS), expenditure on primary health care (SUS), expenditure on medium- and high-complexity health care (SUS), primary public outpatient care, medium-complexity public outpatient care, public nursing assistants and technicians, and public nurse and public physician because they register factor loads higher than 0.5. These factors also present the lowest parameters of uniqueness of less than 0.5.

After the selection of the variables, the standard Z score of the original variables was used to construct the dendrogram below (Figure 2). The dendrogram suggests a cut in four clusters that are detailed in the results of this article according to k-means clustering.

## 3. Results

### 3.1. Health Expenditure

The analysis of the monetary value of public health expenditure “in” the municipalities showed that human resource expenditure increased from the year 2008 to the year 2015 in all regions and in almost all population strata of the country, constant Brazil (BRL) prices of 2015. The term “in” represents municipal expenditure made from all sources of available health resources—federal, state, and own resources managed by the city [15]. The South, Midwest, and Northeast regions stood out the most, growing above 50% on average, considering the exchange rates of 2008 (USD 0.5450) and 2015 (USD 0.3001). In the South, the effect was basically due to the increase in spending in all cities, but especially in those with a population between 20 and 50 thousand inhabitants. The Northeast region followed the same pattern but with one difference: the cities between 500 thousand and 1 million were those responsible for increasing personnel expenditures. In cities above 1 million inhabitants, there was a 20% drop in expenses under this heading (Table 5).

It is important to note that Brazil has 5570 cities, clustered in 26 states plus one Federal District, which are grouped within five geoeconomic regions—North, Northeast, Midwest, Southeast, and South. The Brazilian geoeconomical regions are North (No), which groups the states of Acre (AC), Amapá (AP), Amazonas (AM), Pará (PA), Rondônia (RO), Roraima (RR), and Tocantins (TO); Northeast (NoE), with Alagoas (AL), Bahia (BA), Ceará (CE), Maranhão (MA), Paraíba (PB), Pernambuco (PE), Piauí (PI), Rio Grande do Norte (RN), and Sergipe (SE) states; Midwest (MiW), with the Federal District (DF) and the states of Goiás (GO), Mato Grosso (MT), and Mato Grosso do Sul (MS); Southeast (SoE), with Espírito Santo (ES), Minas Gerais (MG), Rio de Janeiro (RJ) and São Paulo (SP); and South (So), which includes the states of Paraná (PR), Santa Catarina (SC) and Rio Grande do Sul (RS). The North and Northeast regions of Brazil are historically poor regions and have always had low levels of public health services. Southeast and South are rich regions with considerable levels of service offer. The Midwest is a relatively new region, where occupation effectively started in 1950.

Public investment was lower in relation to human resources; this is due to the particularities of the health sector [16]. The production of services is labor intensive, and its operation consumes a good part (in Brazil, on average, 70% [17] of the establishment’s resources [18]. It grew in all regions, except in the Southeast with a 30% drop. The low investment inversion occurs mainly in the 24 largest cities that make up the medium-large stratum and two larger cities within the large group, Campinas (SP state, SoE region) and Curitiba (capital, PR state, therefore, So region) (Table 6).

Despite their population size, medium-large cities show considerable year-to-year fluctuations in their investment expenditures. Some of the highlights are the cities of Sorocaba (SP, SoE), João Pessoa (capital, PB, NoE), and Duque de Caxias (RJ, SoE), which together drastically reduced their investments, with the former dropping from the millions to the thousands. The cities of Ananindeua (PA, No), Uberlândia (MG, SoE), Ribeirão Preto (SP, SoE), Santo André (SP, SoE), and Campo Grande (capital, MS, MiW) also presented considerable decrease. These variations suggest a certain structural constraint to the continuous and consistent expansion of expenditures, where a large part of these municipalities is still sensitive to federal transfers. This is different in the cases of Campinas (SP, SoE) and Curitiba (PR, So), both cities with a population of more than 1 million inhabitants, which are cities with some stability in their own resources but which show a fall in expenditure.

Despite this, some cities increased investment spending, with the expansion of network service. In the medium-large stratum, the cities of Cuiabá (capital, MG, MiW) and Teresina (capital, PI, NoE) and the large cities of Belém (capital, PA, No), Rio de Janeiro (capital, RJ, SoE), and Porto Alegre (capital, RS, So) doubled their expenditure.

The analysis of the health blocks indicates that there was an increase in primary health care spending in all regions of the country. The Southeast, South, and Midwest regions increased more than half of their expenditures. Cities with less than 20,000 inhabitants and large cities were responsible for this increase; the variation of these groups reached 43% and 165%, respectively (Table 7).

Medium- and high-complexity care contracted in almost all regions, except in the Southeast, an effect caused, in part, by cities with less than 20 thousand inhabitants that do not have such a service. However, others consistently increased their average expenditures, especially in cities with a population between 50 thousand and 500 thousand inhabitants. (Table 8).

In this analysis, two types of increase in public health coverage were observed. One linked to the benefits of investment growth in the same direction as medium- and high-complexity care, and another with a rather negative profile, in which investments fell alongside an increase in medium- and high-complexity care spending, suggesting unplanned commercialization of public services.

The Northeast and Midwest regions were the areas where the greatest increase in spending was observed, specifically in the medium-large and large city strata, where investment increased by the same amount as medium- and high-complexity care spending. On the contrary, the Southeast region was the prime location for the expansion of the medium- and high-complexity care with steep drops in investment, especially in the small-middle city strata. The Southeast region was the only one to grow above 50% of the average municipal expenses in medium- and high-complexity care, going from BRL 13 million to BRL 21 million.

However, the expansion of public spending was largely permitted by the growth of the economy in the period between 2006 and 2014 and the increase of fund-to-fund transfers by the federal government. This has reduced some disparities in the distribution of resources. The years analyzed were marked by the growth of public spending, mainly in primary care and consistently in the strata for medium-/high-complexity care. Without this, perhaps, Figure 3 would present greater imbalances than those already evident.

### 3.2. Concentration and Imbalance

The public installed capacity in primary care grew by 8000 units across all regions, with the North and the South growing by 22% and 19%, respectively. Between 2008 and 2015, public supply surpassed the private network in absolute and relative terms, as seen in Table 9. However, the change is more due to private reduction than an increase in public supply. Negative changes in the private sector averaged 50% in almost all regions, with the Midwest and South regional changes being more pronounced (57% and 55%, respectively). Across all primary care units, the participation of the private sector showed a strong exit trend, falling from 49 thousand to 23 thousand in the period.

In fact, the average number of public primary health care units per municipality increased in all regions of the country. The North and Northeast regions increased by 2 points (9.5–11.5 and 10.2–12.1, respectively), while the private sector registered a consistent average reduction of its establishments, falling from 21.9 to 12.

The data suggest that there has been great progress in the universalization of primary care by SUS. Regionalization expanded in the regions and municipalities identified by the service, especially in small towns (0 to 50 thousand inhabitants). There was also a gradual inclusion of medium-sized urban centers (100 thousand to 500 thousand inhabitants) and the maintenance in absolute terms of the network in large cities (over 1 million inhabitants) with the dispersal of the establishments (Table 10).

The private sector shows considerable declines, both in absolute and average numbers of primary facilities in all regions and population strata. In almost all regions, falls were above 50%. The only exception was the North region, which already has a small number of primary care units (963 to 870 establishments), with a reduction of approximately 10%.

As a result, Figure 4 shows that 70% of the municipalities (mostly cities with up to 20 thousand inhabitants) have, on average, six public primary health care establishments. The municipalities with up to 50 thousand inhabitants have, on average, 14 units, revealing a greater regional homogenization (the term “regional homogenization” refers to the regional disparity network offering service) of the public installed capacity in primary health care, with the geographic inequalities in health outcomes significantly reducing. The geographical visualization allows us to assert that for primary health care, there was, in fact, almost complete universalization of the installed capacity. The low level of voids in primary health care establishments across the country is a sign of this. There are few localities with public health care gaps.

The private sector, on the contrary, presents considerable geographic gaps. The sector has, on average, two primary establishments in the small towns and six in the municipalities with up to 50 thousand inhabitants. This is in the context of a sharp increase in private health insurance [19].

The analysis of the evolution of medium-complexity health care in the period (2008–2015) reveals a consistent growth in the installed regional capacity, mainly in the form of emergency care units (UPAs). Table 11 shows that there was an increase of more than 60% in public medium-complexity health care capacity in all regions. The South and North regions registered more significant expansions in the medium-complexity outpatient health care mode.

This increase in public medium-complexity health care facilities mainly took place in cities with less than 20 thousand inhabitants that expanded their installed capacity by 7696 units, and cities between 20,000 and 50,000 inhabitants, with 3674 new units (Table 12).

In the medium-complexity hospital care mode, the Midwest region is the main highlight with a significant increase of 135% (M/hosp), followed by the Northeast and North regions with 58% and 53%, respectively. With respect to high-complexity care, public hospital establishments have strongly decreased their presence in almost all regions [20]. The overall reduction in the high-complexity hospital care mode was 17% (H/hosp) in total. The Northeast and South regions had the highest contractions of 35% and 42%, respectively. The North region was the only one to generate growth of installed capacity in this period. Cities with 500,000 to 1 million inhabitants and those with more than 1 million inhabitants were responsible for reducing the number of high-complexity care hospitals, especially in the Southeast and Northeast regions.

Table 13 shows that the North region had the greatest expansion in the installed capacity of private units for medium-complexity outpatient care, approximately a 60% expansion, followed by the Midwest region with a 10% increase. Similar to the public sector, in absolute terms, small cities with less than 50 thousand inhabitants were the ones that collaborated most in the growth of private installed capacity for medium-complexity care (Table 14).

The same is true in the high-complexity hospital sphere, which, during 2008–2015, also expanded in absolute and relative terms. The national average growth was 31% (80% in the North region and approximately 50% in the Midwest region). The elevated average of private establishments in almost all modalities in medium- and high-complexity care are the result of imbalances in the network. The analysis of the absolute values in the private sector according to population strata verifies that this expansion was highly concentrated in medium and large cities.

In large cities (above 1 million), the average of private medium-complexity outpatient facilities was eight times higher than the average of the public network (about 20 thousand more units). The network of private medium-complexity care hospitals (M/hosp) also grew in the period, although to a lesser degree to those of high complexity. The North and Midwest regions (36% and 29%, respectively) were the highlights. This expansion was concentrated in small (50,000 to 100,000 inhabitants) and medium-sized cities (100,000 to 500,000 inhabitants). As for high-complexity care (H/hosp), the expansion of the private network was even greater, especially in the North and Southeast regions, whose expansion reached 56% and 17%, respectively.

In summary, the public regionalization of medium-complexity outpatient facilities indicates that there has been decentralized growth in many of the country’s cities, except for the medium-large (500,000 to 1 million) and large (over 1 million) strata, thus covering previously undersupplied regions. The private sector expanded its installed capacity in all modes of medium- and high-complexity care, possibly in a move for higher profit margins.

At first glance, the maps (Figure 5 and Figure 6) suggest that the two health care sectors appear to operate in the same localities with a slight reduction in private practice in some regions. However, it must be remembered that the private sector serves only 25% of the Brazilian population, while the public covers 75%. On average, the structures are almost equivalent regionally, with the greatest contrast in the high-complexity care mode, where the public covers most of the country’s territory.

### 3.3. Work in Health

During the analyzed period, SUS expanded human resources across Brazil. One of the constraints of this process was the Family Health Strategy (Estratégia Saúde da Família-ESF, in Portuguese) Program. Between 1998 and 2010, the ESF program grew rapidly from 4000 teams to over 31,600 and was able to expand coverage from 10.6 million to over 100 million registered people [4].

The Family Health Strategy was inspired by the Community Health Agents Program (Programa de Agentes Comunitários de Saúde-PACS, in Portuguese), a community health initiative piloted in rural areas of Ceará during the 1980s. The Family Health Strategy (ESF) was initially developed in parallel with the PACS, gradually replacing it. It was designed to provide first-contact, comprehensive, and whole-person care coordinated with other health services, emphasizing care that takes place within the context of family and community. In the ESF, multiprofessional health teams (composed of a physician, a nurse, a nurse assistant, and four to six community health workers) are organized by geographic regions, with each team providing primary care to approximately 1,000 families (or about 3500 people) [4].

Increased federal transfers and the expansion of private insurance also contributed to the expansion of health jobs. Public and private health sector job position offers increased in all regions and population strata, and the public occupational profile became more homogeneous between 2008 and 2015. The standard deviation of occupation for the four analyzed health worker groups in the public sector was almost 60% lower than in the private sector, where the North, Northeast, and South regions presented greater homogeneity. In both public and private segments, the growth in employment did not significantly alter the relative composition of the supply of physicians, nurses, nursing assistants, and technicians, but a larger difference can be observed between the number of nurses, nursing assistants, and technicians in the private sector, in a movement possibly caused by occupational substitution (Table 15 and Table 16).

Cities of 0 to 100 thousand inhabitants are largely responsible for the growth in all occupations. The increase in the number employed in the three groups of health professionals (n. assist/tech, nurse, and physician) working in the SUS (136 thousand) was much higher than in the private sphere (23 thousand).

Cities from 0 to 50 thousand inhabitants increased the absolute number of SUS physicians by about 31 thousand, while the private sector created only 539 new jobs. The largest absolute variation in the number of SUS physicians (35,000) was mainly concentrated in the medium-sized cities (100,000 to 500,000 inhabitants), while in the private sector, the expansion was concentrated in cities with more than 1 million inhabitants, with 30,000 more employed (Table 17 and Table 18).

In large cities, the increase in the number of SUS physicians, from 105 thousand to 124 thousand, was accompanied by a drop in the average number of these professionals per city, from 7.5 thousand to 6.5 thousand, suggesting an increase in the number of physicians but less geographically concentrated, thus improving geographical distribution. While in the private sector, the increase in the absolute number of occupations of physicians, from 84 thousand to 115 thousand, across the same stratum suggests a higher concentration of professionals, which increased from 6000 to 8200, that is, the public system is expanding supply in all regions of the country, especially in medium-sized cities, while the private sector is more concentrated in large cities.

### 3.4. Cluster Analysis

The SUS clusters reveal a parallel with prevailing regional health theory, according to which the process of regionalization is still in its first steps and facing a serious persistence of gaps in medium- and high-complexity care, in addition to the concentration of full installed capacity in just a few cities. From cluster analysis, we found four groups (Table 19, Table 20, Table 21 and Table 22).

The concentration of installed capacity, which conditions and reflects the access flows to the health network, shows the Southeast region, excluding Espírito Santo, to be extremely dense, along with the cities of Belo Horizonte, São Paulo, and Rio de Janeiro. They invest almost twice the financial resources of the “central cities” cluster, 90% more than the “medium cities” cluster and 40% more than the “small cities” cluster.

The Southeast group’s average spending is BRL 1.5 billion, against BRL 124 million, BRL 31 million, and BRL 5 million in the “central cities”, “medium cities”, and “small towns” groups, respectively. Even with the clearly planned and rational objective of achieving economies of scale, the results indicate that a strong and persistent concentration of the network is still to be found in a small group of cities.

In quantitative terms, the average number of high-complexity outpatient clinics in Belo Horizonte (capital, MG, SoE), São Paulo (capital, SP, SoE), and Rio de Janeiro (capital, RJ, SoE) is four times higher than the central cities group. In this respect, the issue does not necessarily adhere to the logic of the system but rather is disproportionate, because as the technological density of the coverage decreases, the average number and the dispersal of services increases.

Almost every state in Brazil connects to the southeastern region, especially São Paulo, for access to the most complex care services. For myocardial revascularization, only the states of Ceará and Rio Grande do Sul are relatively autonomous. Flowing in the opposite direction and due to extreme isolation, the inhabitants of Rondônia travel to all regions of the country seeking treatment—the flows go from the State of Pará (No), through Rio Grande do Norte (NoE), Goiás (MiW) and São Paulo (SoE) and arrive as far away as Paraná (So) [21].

Moreover, alongside the significant concentration of high-complexity care, the heterogeneity of SUS installed capacity is yet another point that needs to be improved. The k-means reveals that 3243 municipalities did not belong to any group at all, suggesting strong heterogeneity in the distribution of the SUS public offering capacity.

## 4. Conclusions

In the 30 years of the SUS, the system has been able to almost universalize primary care and consistently advance medium-complexity care. The installed physical capacity and the number of professionals have increased in almost all regions and population strata. The need for greater federal and state participation is a challenge to be met since the regional balance of health supply depends on their actions [22]. The other challenge is overcoming the Fiscal Responsibility Law (FRL); more flexibility for the health sector would contribute to the greater degree of freedom of the public sector to allocate human resources [19]. The law blocks increase in human resource public spending levels in states and municipalities.

The FRL [23], in its section relating to health, limits the expansion of the public network by requiring that human resource expenditure does not exceed 54% of total spending. Therefore, whenever public health policy attempts to increase coverage by continuing to expand service delivery, the FRL obliges local managers to include private participation in public services [4]. The FRL requires most municipalities to direct their expenditure to services provided by third parties, without guaranteeing that these alternatives align any better with SUS precepts [24].

In macro terms, deficiencies in health coverage are due to the segmented public/private mix model [25], public underfunding [26], and delayed regionalization [27], all taking place in a scenario where economic and social stability is a prerequisite for the development of universal services [28]. As a result, improvements in concentration levels of health care and the heterogeneity of installed capacity are dependent on the conditions of this background.

Finally, the study did not analyze the actual health status of the Brazilian population; rather, it analyzed the installed health capacity offered in this prosperous period. In particular, we sought to analyze the dynamics of occupation of health gaps between the public and private sectors. This is the main limitation of the study.

## Figures and Tables

**Figure 1 healthcare-09-01380-f001:**
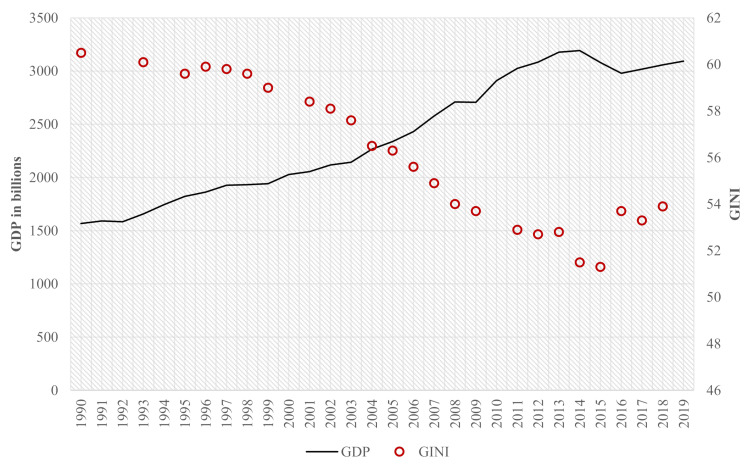
Growth and equality. Source: World Bank. GDP, PPP (constant 2011 international USD), [https://databank.worldbank.org/source/world-development-indicators (accessed on 10 August 2019)]. Elaboration: by authors.

**Figure 2 healthcare-09-01380-f002:**
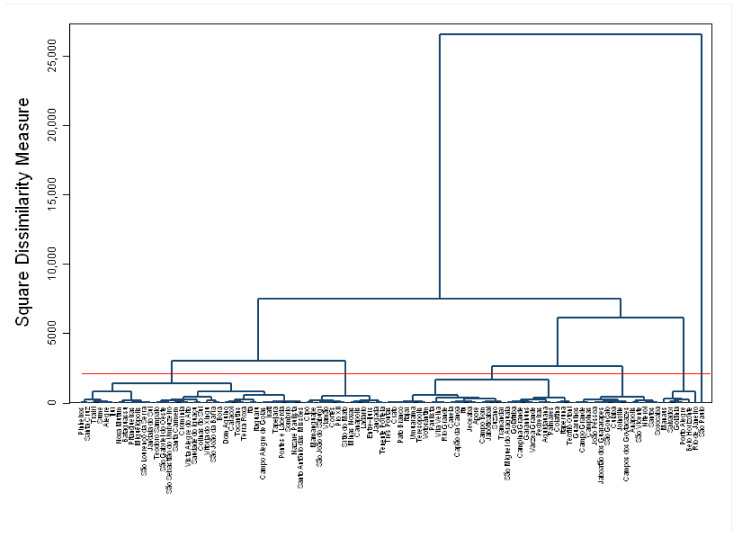
Dendrogram for Ward’s cluster analysis. The color of the dendrogram line is navy and the cut line is red. Source: Share % of Tax Revenue and Constitutional and Legal Transfers in Total Revenue of the Municipality (excluding deductions); Population Census 2010 and 2015 of the Brazilian Institute of Geography and Statistics (Censo Demográfico 2010 e 2015 do Instituto Brasileiro de Geografia e Estatística-IBGE); Information System on Public Health Budgets–ISPHB (Sistema de Informação sobre Orçamentos Públicos em Saúde-SIOPS); Ministry of Health, National Registry of Health Establishments (Cadastro Nacional dos Estabelecimentos de Saúde do Brasil-CNES); National Occupation Classification (Classificação Nacional de Ocupação-CBO); National Agency for Supplementary Health (Agência Nacional de Saúde Suplementar-ANS). Production: by the authors.

**Figure 3 healthcare-09-01380-f003:**
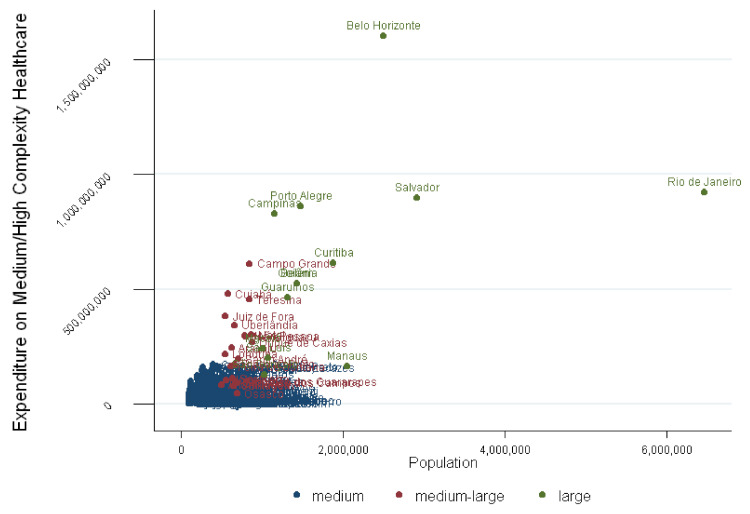
Expenditure on medium/high-complexity health care (SUS) by 272 bigger municipalities, 2015. Note: São Paulo is outlier; the city has higher expenditure than all other cities. Scores colors are as follows: medium population in navy; medium-large population in maroon; large population in green. Source: Information System on Public Health Budgets–ISPHB (Sistema de Informação sobre Orçamentos Públicos em Saúde-SIOPS), Ministry of Health. Production: by the authors.

**Figure 4 healthcare-09-01380-f004:**
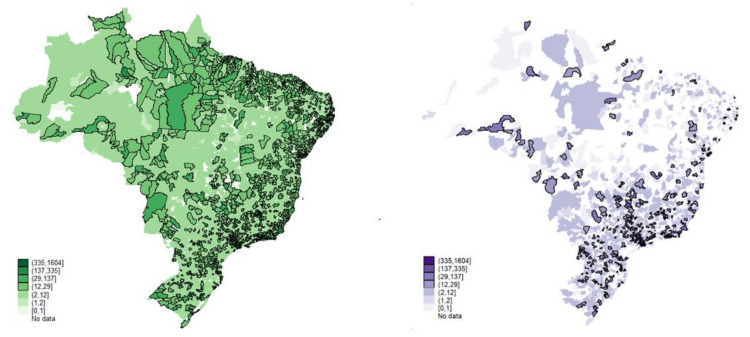
Primary public outpatient care (green-left) and primary private outpatient care, 2015. Source: Ministry of Health, National Registry of Health Establishments (Cadastro Nacional dos Estabelecimentos de Saúde do Brasil-CNES); National Agency for Supplementary Health (Agência Nacional de Saúde Suplementar-ANS). Production: by the authors.

**Figure 5 healthcare-09-01380-f005:**
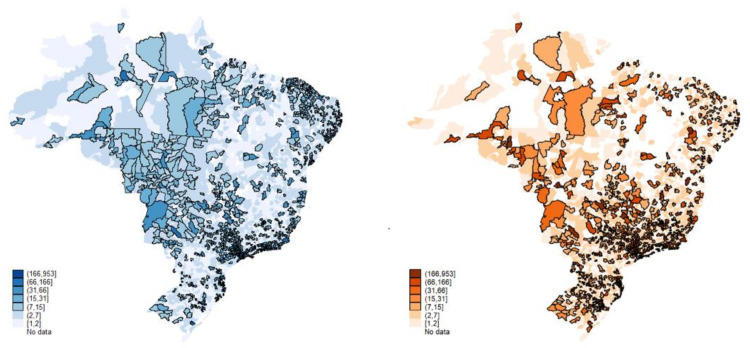
Medium-complexity public outpatient care (blue-left) and medium-complexity private outpatient Care, 2015. Source: Ministry of Health, National Registry of Health Establishments (Cadastro Nacional dos Estabelecimentos de Saúde do Brasil-CNES); National Agency for Supplementary Health (Agência Nacional de Saúde Suplementar-ANS). Production: by the authors.

**Figure 6 healthcare-09-01380-f006:**
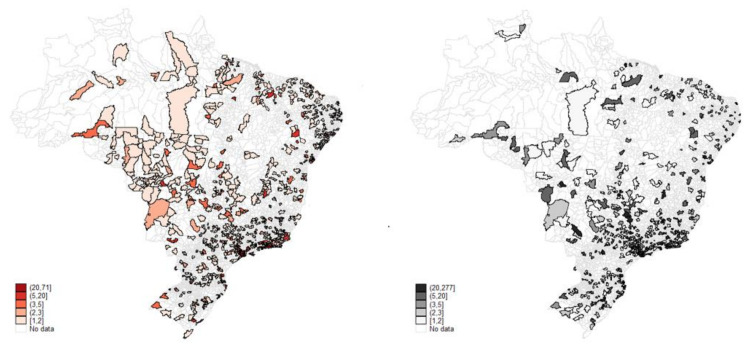
Higher complexity public outpatient care and higher complexity private outpatient care, 2015. Source: Ministry of Health, National Registry of Health Establishments (Cadastro Nacional dos Estabelecimentos de Saúde do Brasil-CNES); National Agency for Supplementary Health (Agência Nacional de Saúde Suplementar-ANS). Production: by the authors.

**Table 1 healthcare-09-01380-t001:** Observation (available), absent (unavailable), and outlier (by municipality).

N	Variable	Absent	Outlier	Observations	Source
1	Population			5568	IBGE
2	Population by age group (% of those over 59 years of age)			5568	IBGE
3	Tax Revenue/Total Revenue (SUS)	1		5567	SIOPS
4	Percentage of Constitution Transfer Taxes Receipt in relation to Total Municipality Receipt * (SUS)	1		5567	SIOPS
5	Transfer Central Government (Union) to the SUS/Transfer Central Government (Union)	1		5567	SIOPS
6	Applied Own Resources (Constitutional Amendment 29)	1		5567	SIOPS
7	Resources (*n* = 3 + 4 + 5 + 6) per inhabitants	1		5567	SIOPS
8	Human Resource Expenditure (SUS)	19	301	5248	SIOPS
9	Investment (SUS)	122	406	5040	SIOPS
10	Expenditure on Primary Health care (SUS)	315	199	5054	SIOPS
11	Expenditure on Medium- and High-Complexity Health care (SUS)	1464	328	3776	SIOPS
12	Primary Public Outpatient Care	8		5561	CNES
13	Medium-Complexity Public Outpatient Care	166		5403	CNES
14	High-Complexity Public Outpatient Care	4748		820	CNES
15	Medium-Complexity Public Hospital Care	4227		1291	CNES
16	High-Complexity Public Hospital Care	5440		128	CNES
17	Public Nursing Assistants and Technicians	227		5341	CBO
18	Public Nurse	215		5353	CBO
19	Public Doctor (physician)	217		5351	CBO
20	Primary Private Outpatient Care	3542		2027	CNES/ANS
21	Medium-Complexity Private Outpatient Care	2616		2953	CNES/ANS
22	High-Complexity Private Outpatient Care	4997		571	CNES/ANS
23	Medium-Complexity Private Hospital Care	5003		565	CNES/ANS
24	High-Complexity Private Hospital Care	5371		197	CNES/ANS
25	Private Nursing Assistants and Technicians	4721		847	CBO
26	Private Nurse	4804		764	CBO
27	Private Doctor (physician)	3500		2068	CBO

* Share % of Tax Revenue and Constitutional and Legal Transfers in Total Revenue of the Municipality (excluding deductions); Population Census 2010 and 2015 of the Brazilian Institute of Geography and Statistics (Censo Demográfico 2010 e 2015 do Instituto Brasileiro de Geografia e Estatística-IBGE); Information System on Public Health Budgets–ISPHB (Sistema de Informação sobre Orçamentos Públicos em Saúde-SIOPS); Ministry of Health, National Registry of Health Establishments (Cadastro Nacional dos Estabelecimentos de Saúde do Brasil-CNES); National Occupation Classification (Classificação Nacional de Ocupação-CBO); National Agency for Supplementary Health (Agência Nacional de Saúde Suplementar-ANS). Production: by the authors.

**Table 2 healthcare-09-01380-t002:** Variables to cluster analysis.

N	Variable
1	Population
2	Population by age group (% of those over 59 years of age)
3	Tax Revenue/Total Revenue (SUS)
4	Percentage of Constitution Transfer Taxes Receipt in relation to Total Municipality Receipt * (SUS)
5	Transfer Central Government (Union) to the SUS/Transfer Central Government (Union)
6	Applied Own Resources (Constitutional Amendment 29)
7	Resources (*n* = 3 + 4 + 5 + 6) per inhabitants
8	Human Resource Expenditure (SUS)
9	Expenditure on Primary Health care (SUS)
10	Expenditure on Medium- and High-Complexity Health care (SUS)
11	Primary Public Outpatient Care
12	Medium-Complexity Public Outpatient Care
13	Public Nursing Assistants and Technicians
14	Public Nurse
15	Public Doctor (physician)

Source: Share % of Tax Revenue and Constitutional and Legal Transfers in Total Revenue of the Municipality (excluding deductions); * It is the part of the municipality’s resources that are not its own resources, coming from the central and middle government (federal and state); Population Census 2010 and 2015 of the Brazilian Institute of Geography and Statistics (Censo Demográfico 2010 e 2015 do Instituto Brasileiro de Geografia e Estatística-IBGE); Information System on Public Health Budgets–ISPHB (Sistema de Informação sobre Orçamentos Públicos em Saúde-SIOPS); Ministry of Health, National Registry of Health Establishments (Cadastro Nacional dos Estabelecimentos de Saúde do Brasil-CNES); National Occupation Classification (Classificação Nacional de Ocupação-CBO); National Agency for Supplementary Health (Agência Nacional de Saúde Suplementar-ANS). Production: by the authors.

**Table 3 healthcare-09-01380-t003:** Factor analysis.

Factor	Eigenvalue	Difference	Proportion	Cumulative
Factor1	7.9466	6.4164	0.7476	0.7476
Factor2	1.5302	0.7870	0.1439	0.8915
Factor3	0.7432	0.1125	0.0699	0.9614
Factor4	0.6307	0.5039	0.0593	1.0208
Factor5	0.1268	0.0630	0.0119	1.0327
Factor6	0.0638	0.0312	0.0060	1.0387
Factor7	0.0326	0.0323	0.0031	1.0418
Factor8	0.0003	0.0045	0.0000	1.0418
Factor9	−0.0042	0.0079	−0.0004	1.0414
Factor10	−0.0121	0.0030	−0.0011	1.0403
Factor11	−0.0151	0.0132	−0.0014	1.0388
Factor12	−0.0283	0.0104	−0.0027	1.0362
Factor13	−0.0387	0.0576	−0.0036	1.0325
Factor14	−0.0963	0.1532	−0.0091	1.0235
Factor15	−0.2495		−0.0235	1.0000

Source: Share % of Tax Revenue and Constitutional and Legal Transfers in Total Revenue of the Municipality (excluding deductions); Population Census 2010 and 2015 of the Brazilian Institute of Geography and Statistics (Censo Demográfico 2010 e 2015 do Instituto Brasileiro de Geografia e Estatística-IBGE); Information System on Public Health Budgets–ISPHB (Sistema de Informação sobre Orçamentos Públicos em Saúde-SIOPS); Ministry of Health, National Registry of Health Establishments (Cadastro Nacional dos Estabelecimentos de Saúde do Brasil-CNES); National Occupation Classification (Classificação Nacional de Ocupação-CBO); National Agency for Supplementary Health (Agência Nacional de Saúde Suplementar-ANS). Production: by the authors.

**Table 4 healthcare-09-01380-t004:** Factor analysis.

Variable	Factor1	Factor2	Factor3	Factor4	Factor5	Factor6	Factor7	Factor8	Uniqueness
Population	**0.97**	0.05	−0.18	0.02	−0.14	−0.02	0.02	0.00	0.01
Population by age group (% of those over 59 years of age)	−0.07	0.50	0.00	−0.04	0.08	−0.03	0.09	0.00	0.73
Tax Revenue/Total Revenue (SUS)	0.39	0.08	0.42	0.29	−0.09	0.00	−0.05	−0.01	0.57
Percentage of Constitution Transfer Taxes Receipt in relation to Total Municipality Receipt * (SUS)	0.43	−0.53	0.31	0.14	0.12	−0.01	0.06	0.00	0.40
Transfer Central Government (Union) to the SUS/Transfer Central Government (Union)	−0.09	0.76	0.03	0.04	−0.02	−0.09	−0.03	0.00	0.40
Applied Own Resources (Constitutional Amendment 29)	0.07	0.13	0.29	0.22	−0.07	0.14	0.01	0.00	0.82
Resources (*n* = 3 + 4 + 5 + 6) per inhabitants	0.04	0.56	0.24	0.13	0.08	0.03	0.04	0.00	0.60
Human Resource Expenditure (SUS)	**0.97**	0.07	−0.01	0.02	−0.06	−0.02	−0.07	0.01	0.04
Expenditure on Primary Health care (SUS)	**0.80**	0.11	−0.44	0.28	0.01	0.08	0.02	0.00	0.07
Expenditure on Medium- and High-Complexity Health care (SUS)	**0.94**	0.09	−0.18	0.08	0.17	0.05	−0.05	0.00	0.03
Primary Public Outpatient Care	**0.89**	−0.20	0.07	0.16	−0.05	−0.12	0.04	0.00	0.11
Medium-Complexity Public Outpatient Care	**0.95**	−0.03	−0.03	0.20	0.07	−0.06	0.01	−0.01	0.05
Public Nursing Assistants and Technicians	**0.88**	0.04	0.24	−0.36	0.04	−0.02	−0.02	0.00	0.03
Public Nurse	**0.92**	0.08	0.02	−0.31	−0.13	0.07	0.08	0.00	0.02
Public Doctor (physician)	**0.92**	0.08	0.12	−0.28	0.09	0.04	−0.03	−0.01	0.04

Source: * Share % of Tax Revenue and Constitutional and Legal Transfers in Total Revenue of the Municipality (excluding deductions); Population Census 2010 and 2015 of the Brazilian Institute of Geography and Statistics (Censo Demográfico 2010 e 2015 do Instituto Brasileiro de Geografia e Estatística-IBGE); Information System on Public Health Budgets–ISPHB (Sistema de Informação sobre Orçamentos Públicos em Saúde-SIOPS); Ministry of Health, National Registry of Health Establishments (Cadastro Nacional dos Estabelecimentos de Saúde do Brasil-CNES); National Occupation Classification (Classificação Nacional de Ocupação-CBO); National Agency for Supplementary Health (Agência Nacional de Saúde Suplementar-ANS). Production: by the authors. Bold: From Factor 1, the measurement of sampling adequacy (MSA) applied was 0.80 to select the original variables.

**Table 5 healthcare-09-01380-t005:** Human resource public expenditure and investment (SUS) by region (municipal average).

Region	2008		2015	
	Human Resource	Investment	Human Resource	Investment
North	6,287,995.51	386,747.57	8,865,348.43	522,688.84
Northeast	5,122,279.46	373,378.44	7,657,237.77	443,252.25
Southeast	11,044,496.33	896,930.49	13,731,783.04	635,463.90
South	4,128,870.57	314,157.00	7,318,001.24	392,144.97
Midwest	5,687,828.45	419,646.09	8,550,003.18	523,722.04
Total	6,829,583.00	526,474.44	9,562,787.52	502,806.45

Production: by the authors using the calculator of Brazil’s Central Bank (Bacen). Broad Consumer Price Index (IPCA) of Brazilian Institute of Geography and Statistics (IBGE), cumulative index of 1.5577600 (2008–2015). Constant Brazil (BRL) prices of 2015. The exchange rates of 2008 (USD 0.5450) and 2015 (USD 0.3001).

**Table 6 healthcare-09-01380-t006:** Human resource public expenditure and investment (SUS) by population stratum (municipal average).

Classification	Stratum	2008		2015	
		Human Resource	Investment	Human Resource	Investment
Very small	Pop. ≤ 20,000	1,647,784.97	194,628.29	2,416,684.74	198,923.42
Small	20,000 < pop. ≤ 50,000	5,163,202.04	431,864.55	7,739,382.61	518,680.74
Small-medium	50,000 < pop. ≤ 100,000	13,009,326.44	964,336.13	18,887,393.73	1,104,100.30
Medium	100,000 < pop. ≤ 500,000	39,036,283.13	2,883,016.71	58,879,731.52	2,489,403.92
Medium-large	500,000 < pop. ≤ 1 million	155,469,020.26	12,141,730.53	224,991,303.48	8,165,209.30
Large	Pop. > 1 million	659,658,957.56	36,822,941.85	530,037,004.91	18,555,634.18
Total		6,829,583.00	526,474.44	9,562,787.52	502,806.45

Production: by the authors using the calculator of Brazil’s Central Bank (Bacen). Broad Consumer Price Index (IPCA) of Brazilian Institute of Geography and Statistics (IBGE), cumulative index of 1.5577600 (2008–2015). Constant BRL prices of 2015. The exchange rates of 2008 (USD 0.5450) and 2015 (USD 0.3001).

**Table 7 healthcare-09-01380-t007:** Expenditure on primary and medium-/high-complexity health care (SUS) by region (municipal average).

Region	2008		2015	
	Primary	Medium/High	Primary	Medium/High
North	4,278,745.21	9,356,899.64	5,941,717.70	8,644,777.30
Northeast	4,278,458.18	9,198,694.66	4,625,240.42	6,568,575.50
Southeast	5,694,131.73	13,372,184.30	10,869,535.32	21,967,488.92
South	3,754,707.80	12,680,853.50	6,742,064.67	10,024,791.34
Midwest	4,019,066.12	10,783,831.80	6,141,224.23	10,320,658.85
Total	4,572,140.13	11,479,861.12	7,166,177.86	12,392,791.87

Production: by the authors using the calculator of Brazil’s Central Bank (Bacen) to the value actualization of in real terms. Broad Consumer Price Index (IPCA) of Brazilian Institute of Geography and Statistics (IBGE), cumulative index of 1.5577600 (2008–2015). Constant BRL prices of 2015. The exchange rates of 2008 (USD 0.5450) and 2015 (USD 0.3001).

**Table 8 healthcare-09-01380-t008:** Expenditure on primary and medium-/high-complexity health care (SUS) by population stratum (municipal average).

Classification	Stratum	2008		2015	
		Primary	Medium/High	Primary	Medium/High
Very small	Pop. ≤ 20,000	1,981,679.84	799,566.17	2,832,844.78	836,145.48
Small	20,000 < pop. ≤ 50,000	5,275,363.18	2,360,388.42	6,080,622.29	3,432,995.93
Small-medium	50,000 < pop. ≤ 100,000	9,245,552.84	9,587,156.34	12,164,499.95	15,360,956.73
Medium	100,000 < pop. ≤ 500,000	24,871,977.72	29,802,515.40	29,933,352.60	47,274,849.48
Medium-large	500,000 < pop. ≤ 1 million	73,779,468.89	162,266,960.28	76,283,383.17	243,990,809.82
Large	Pop. > 1 million	135,042,042.73	590,048,301.64	357,684,973.87	821,740,560.90
Total		4,572,140.13	11,479,861.12	7,166,177.86	12,392,791.87

Production: by the authors using the calculator of Brazil’s Central Bank (Bacen) to the value actualization of in real terms. Broad Consumer Price Index (IPCA) of Brazilian Institute of Geography and Statistics (IBGE), cumulative index of 1.5577600 (2008–2015). Constant BRL prices of 2015. The exchange rates of 2008 (USD 0.5450) and 2015 (USD 0.3001).

**Table 9 healthcare-09-01380-t009:** Primary outpatient care by region.

Region	2008		2015	
	Public	Private	Public	Private
North	4260	963	5179	870
Northeast	18,030	7001	21,111	3287
Southeast	17,218	25,453	18,976	11,858
South	8017	12,134	9573	5500
Midwest	3340	4162	3935	1777
Total	50,865	49,713	58,774	23,292

Source: Ministry of Health, National Registry of Health Establishments (Cadastro Nacional dos Estabelecimentos de Saúde do Brasil-CNES). Production: by the authors.

**Table 10 healthcare-09-01380-t010:** Primary outpatient care by population stratum.

Classification	Stratum	2008		2015	
		Public	Private	Public	Private
Very small	Pop. ≤ 20,000	17,451	3509	22,871	1766
Small	20,000 < pop. ≤ 50,000	12,494	7080	14,567	3633
Small-medium	50,000 < pop. ≤ 100,000	7060	8022	7526	3567
Medium	100,000 < pop. ≤ 500,000	8952	16,149	9123	6471
Medium-large	500,000 < pop. ≤ 1 million	2168	6236	1854	3162
Large	Pop. > 1 million	2740	8717	2833	4693
Total		50,865	49,713	58,774	23,292

Source: Ministry of Health, National Registry of Health Establishments (Cadastro Nacional dos Estabelecimentos de Saúde do Brasil-CNES). Production: by the authors.

**Table 11 healthcare-09-01380-t011:** Medium and higher public outpatient and hospital care by region.

Region	2008				2015			
	M/out	H/out	M/hosp	H/hosp	M/out	H/out	M/hosp	H/hosp
North	1032	54	120	13	2321	95	184	15
Northeast	5340	344	480	63	10,455	458	760	41
Southeast	8389	685	296	102	13,187	895	410	102
South	1891	150	55	19	4930	248	72	11
Midwest	1637	102	101	21	3435	191	237	13
Total	18,289	1335	1052	218	34,328	1887	1663	182

Note: medium outpatient (M/out); higher outpatient (H/out); medium hospital (M/hosp); higher hospital (H/hosp). Source: Ministry of Health, National Registry of Health Establishments (Cadastro Nacional dos Estabelecimentos de Saúde do Brasil-CNES). Production: by the authors.

**Table 12 healthcare-09-01380-t012:** Medium and higher public outpatient and hospital care by population stratum.

Classification	Stratum	2008				2015			
		M/out	H/out	M/hosp	H/hosp	M/out	H/out	M/hosp	H/hosp
Very small	Pop. ≤ 20,000	2847	79	268	6	10,543	223	669	11
Small	20,000 < pop. ≤ 50,000	3357	151	218	9	7031	351	388	17
Small-medium	50,000 < pop. ≤ 100,000	3013	222	106	17	4946	344	135	21
Medium	100,000 < pop. ≤ 500,000	5312	443	151	60	7131	574	205	67
Medium-large	500,000 < pop. ≤ 1 million	1338	134	108	35	1520	127	113	26
Large	Pop. > 1 million	2422	306	201	91	3157	268	153	40
Total		18,289	1335	1052	218	34,328	1887	1663	182

Note: medium outpatient (M/out); higher outpatient (H/out); medium hospital (M/hosp); higher hospital (H/hosp). Source: Ministry of Health, National Registry of Health Establishments (Cadastro Nacional dos Estabelecimentos de Saúde do Brasil-CNES). Production: by the authors.

**Table 13 healthcare-09-01380-t013:** Medium and higher private outpatient and hospital care by stratum.

Region	2008				2015			
	Med/Out	High/Out	Med/Hosp	High/Hosp	Med/Out	High/Out	Med/Hosp	High/Hosp
North	1881	113	99	14	3005	175	135	25
Northeast	13,962	550	534	87	14,224	685	669	120
Southeast	47,996	1849	676	266	39,256	2041	775	346
South	15,920	500	166	76	14,431	661	194	82
Midwest	4635	292	251	54	5086	392	324	80
Total	84,394	3304	1726	497	76,002	3954	2097	653

Note: medium outpatient (med/out); higher outpatient (high/out); medium hospital (med/hosp); higher hospital (high/hosp). Ministry of Health, National Registry of Health Establishments (Cadastro Nacional dos Estabelecimentos de Saúde do Brasil-CNES); National Occupation Classification (Classificação Nacional de Ocupação-CBO); National Agency for Supplementary Health (Agência Nacional de Saúde Suplementar-ANS). Production: by the authors.

**Table 14 healthcare-09-01380-t014:** Medium and higher private outpatient and hospital care by population stratum.

Classification	Stratum	2008				2015			
		M/out	H/out	M/hosp	H/hosp	M/out	H/out	M/hosp	H/hosp
Very small	Pop. ≤ 20,000	2953	32	59	0	4188	59	78	2
Small	20,000 < pop. ≤ 50,000	6169	98	152	15	7644	220	201	24
Small-medium	50,000 < pop. ≤ 100,000	10,563	331	201	38	10,250	617	293	56
Medium	100,000 < pop. ≤ 500,000	27,034	1176	393	198	21,994	1414	529	243
Medium-large	500,000 < pop. ≤ 1 million	10,704	598	289	89	8754	672	363	115
Large	Pop. > 1 million	26,971	1069	632	157	23,172	972	633	213
Total		84,394	3304	1726	497	76,002	3954	2097	653

Note: medium outpatient (med/out); higher outpatient (high/out); medium hospital (med/hosp); higher hospital (high/hosp). Ministry of Health, National Registry of Health Establishments (Cadastro Nacional dos Estabelecimentos de Saúde do Brasil-CNES); National Occupation Classification (Classificação Nacional de Ocupação-CBO); National Agency for Supplementary Health (Agência Nacional de Saúde Suplementar-ANS). Production: by the authors.

**Table 15 healthcare-09-01380-t015:** Health professionals by region, 2008.

Region	Public			Private		
	n.aux/Tech	Nurse	Physician	n.aux/Tech	Nurse	Physician
North	9961	5524	12,656	277	100	2401
Northeast	39,250	24,858	77,392	1531	1422	27,414
Southeast	52,300	37,424	184,952	13,048	6750	126,020
South	15,675	8937	42,241	2282	925	29,053
Midwest	12,886	4748	24,477	1006	257	8853
Total	130,072	81,491	341,718	18,144	9454	193,741

Note: auxiliary nurses and technicians (n.aux/tech). Production: Ministry of Health, National Registry of Health Establishments (Cadastro Nacional dos Estabelecimentos de Saúde do Brasil-CNES); National Occupation Classification (Classificação Nacional de Ocupação-CBO); National Agency for Supplementary Health (Agência Nacional de Saúde Suplementar-ANS). Production: by the authors.

**Table 16 healthcare-09-01380-t016:** Health professionals by region, 2015.

Region	Public			Private		
	n.aux/Tech	Nurse	Physician	n.aux/Tech	Nurse	Physician
North	20,918	10,233	16,497	1649	414	4883
Northeast	66,055	43,278	87,345	4939	2255	35,441
Southeast	119,301	73,229	242,125	42,290	18,843	169,874
South	44,401	22,425	73,404	7209	2474	51,468
Midwest	24,916	10,818	34,419	2065	679	13,305
Total	275,591	159,983	453,790	58,152	24,665	274,971

Note: auxiliary nurses and technicians (n.aux/tech). Ministry of Health, National Registry of Health Establishments (Cadastro Nacional dos Estabelecimentos de Saúde do Brasil-CNES); National Occupation Classification (Classificação Nacional de Ocupação-CBO); National Agency for Supplementary Health (Agência Nacional de Saúde Suplementar-ANS). Production: by the authors.

**Table 17 healthcare-09-01380-t017:** Health professionals by population stratum, 2008.

Classification	Stratum	Public			Private		
		n.aux/Tech	Nurse	Physician	n.aux/Tech	Nurse	Physician
Very small	Pop. ≤ 20,000	18,851	10,425	20,616	40	24	1623
Small	20,000 < pop. ≤ 50,000	18,200	10,347	29,126	230	112	6874
Small-medium	50,000 < pop. ≤ 100,000	15,271	8075	37,442	725	292	13,820
Medium	100,000 < pop. ≤ 500,000	32,252	18,522	96,492	4096	1591	55,614
Medium-large	500,000 < pop. ≤ 1 million	15,878	9646	52,484	2832	1390	30,825
Large	Pop. > 1 million	29,620	24,476	105,558	10,221	6045	84,985
Total		130,072	81,491	341,718	18,144	9454	193,741

Note: auxiliary nurses and technicians (n.aux/tech). Ministry of Health, National Registry of Health Establishments (Cadastro Nacional dos Estabelecimentos de Saúde do Brasil-CNES); National Occupation Classification (Classificação Nacional de Ocupação-CBO); National Agency for Supplementary Health (Agência Nacional de Saúde Suplementar-ANS). Production: by the authors.

**Table 18 healthcare-09-01380-t018:** Health professionals by population stratum, 2015.

Classification	Stratum	Public			Private		
		n.aux/Tech	Nurse	Physician	n.aux/Tech	Nurse	Physician
Very small	Pop. ≤ 20,000	34,812	23,455	34,643	281	183	3575
Small	20,000 < pop. ≤ 50,000	36,351	21,925	46,520	1119	492	12,930
Small-medium	50,000 < pop. ≤ 100,000	34,807	18,344	53,879	3540	1407	23,721
Medium	100,000 < pop. ≤ 500,000	74,759	38,236	132,384	15,231	5420	80,042
Medium-large	500,000 < pop. ≤ 1 million	34,138	17,603	62,092	7088	2608	39,520
Large	Pop. > 1 million	60,724	40,420	124,272	30,893	14,555	115,183
Total		275,591	159,983	453,790	58,152	24,665	274,971

Note: auxiliary nurses and technicians (n.aux/tech). Ministry of Health, National Registry of Health Establishments (Cadastro Nacional dos Estabelecimentos de Saúde do Brasil-CNES); National Occupation Classification (Classificação Nacional de Ocupação-CBO); National Agency for Supplementary Health (Agência Nacional de Saúde Suplementar-ANS). Production: by the authors.

**Table 19 healthcare-09-01380-t019:** Southeast cluster.

County	Municipality
1	Belo Horizonte
2	Rio de Janeiro
3	São Paulo

Note: k-means clustering methodology. Source: Share % of Tax Revenue and Constitutional and Legal Transfers in Total Revenue of the Municipality (excluding deductions); Population Census 2010 and 2015 of the Brazilian Institute of Geography and Statistics (Censo Demográfico 2010 e 2015 do Instituto Brasileiro de Geografia e Estatística-IBGE); Information System on Public Health Budgets–ISPHB (Sistema de Informação sobre Orçamentos Públicos em Saúde-SIOPS); Ministry of Health, National Registry of Health Establishments (Cadastro Nacional dos Estabelecimentos de Saúde do Brasil-CNES); National Occupation Classification (Classificação Nacional de Ocupação-CBO); National Agency for Supplementary Health (Agência Nacional de Saúde Suplementar-ANS). Production: by the authors.

**Table 20 healthcare-09-01380-t020:** Central cities cluster.

County	Municipality
1	Guarulhos
2	Natal
3	João Pessoa
4	São Luís
5	Belém
6	Contagem
7	Goiânia
8	Teresina
9	Salvador
10	Campos dos Goytacazes
11	Porto Alegre
12	Curitiba
13	Campo Grande
14	Maceió
15	Aracaju
16	Cuiabá
17	Joinville
18	Manaus
19	Campinas

Note: k-means clustering methodology. Source: Share % of Tax Revenue and Constitutional and Legal Transfers in Total Revenue of the Municipality (excluding deductions); Population Census 2010 and 2015 of the Brazilian Institute of Geography and Statistics (Censo Demográfico 2010 e 2015 do Instituto Brasileiro de Geografia e Estatística-IBGE); Information System on Public Health Budgets–ISPHB (Sistema de Informação sobre Orçamentos Públicos em Saúde-SIOPS); Ministry of Health, National Registry of Health Establishments (Cadastro Nacional dos Estabelecimentos de Saúde do Brasil-CNES); National Occupation Classification (Classificação Nacional de Ocupação-CBO); National Agency for Supplementary Health (Agência Nacional de Saúde Suplementar-ANS). Production: by the authors.

**Table 21 healthcare-09-01380-t021:** Medium cities cluster.

County	Municipality
1	Ananindeua
2	Passo Fundo
3	Suzano
4	Apucarana
5	Divinópolis
6	Ilhéus
7	Cataguases
8	Marabá
9	Concórdia
10	Uberlândia
11	Itaguaí
12	Jaú
13	Araçatuba
14	Osasco
15	Carapicuíba
16	Tatuí
17	Vitória
18	Itatiba
19	Sete Lagoas
20	Gravataí
⋮	⋮
161	Várzea Grande

Note: k-means clustering methodology. Source: Share % of Tax Revenue and Constitutional and Legal Transfers in Total Revenue of the Municipality (excluding deductions); Population Census 2010 and 2015 of the Brazilian Institute of Geography and Statistics (Censo Demográfico 2010 e 2015 do Instituto Brasileiro de Geografia e Estatística-IBGE); Information System on Public Health Budgets–ISPHB (Sistema de Informação sobre Orçamentos Públicos em Saúde-SIOPS); Ministry of Health, National Registry of Health Establishments (Cadastro Nacional dos Estabelecimentos de Saúde do Brasil-CNES); National Occupation Classification (Classificação Nacional de Ocupação-CBO); National Agency for Supplementary Health (Agência Nacional de Saúde Suplementar-ANS). Production: by the authors.

**Table 22 healthcare-09-01380-t022:** Small cities cluster.

County	Municipality
1	Pacatuba
2	Charqueadas
3	Paraíso do Tocantins
4	Ourolândia
5	Santa Cruz do Piauí
6	São José do Seridó
7	Buenópolis
8	Rio do Fogo
9	Três Marias
10	Maués
11	Quipapá
12	Laje
13	Rondon do Pará
14	Fernandópolis
15	Pedro Canário
16	Santo Antônio de Posse
17	Ourilândia do Norte
18	Alcobaça
19	Jaíba
20	Curral de Dentro
⋮	⋮
2144	Ninheira

Note: k-means clustering methodology. Source: Share % of Tax Revenue and Constitutional and Legal Transfers in Total Revenue of the Municipality (excluding deductions); Population Census 2010 and 2015 of the Brazilian Institute of Geography and Statistics (Censo Demográfico 2010 e 2015 do Instituto Brasileiro de Geografia e Estatística-IBGE); Information System on Public Health Budgets–ISPHB (Sistema de Informação sobre Orçamentos Públicos em Saúde-SIOPS); Ministry of Health, National Registry of Health Establishments (Cadastro Nacional dos Estabelecimentos de Saúde do Brasil-CNES); National Occupation Classification (Classificação Nacional de Ocupação-CBO); National Agency for Supplementary Health (Agência Nacional de Saúde Suplementar-ANS). Production: by the authors.

## Data Availability

The data presented in this study are public information, available in the Ministry of Health, specifically in the Hospital Information System (SIH-SUS) and Outpatient Information System (SIA-SUS).

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
