# Peer review of "The Regionalization Process for Universal Health Coverage in Brazil (2008–2015)"

_healthcare, 2021, doi:10.3390/healthcare9101380_

Round 1

Reviewer 1 Report

I  commend the authors for their study title, “The Regionalization Process for Universal Health Coverage in Brazil (2008-2015)”

This is a resubmission of a manuscript I had previously reviewed. The authors have presented a report on the unique experience of almost a decade-long public health policy implementation in Brazil. This reviewer believes the report could be very important at the local, regional, and national levels for the public, private, and all other stakeholders in Brazil’s Health System.

This reviewer is not able to determine the design of the study,  the research question(s), the sampling method, the sampling unit, and the outcome variable (s,) and the independent variables that significantly influenced the outcome for this study. The authors also do not provide counterfactual evidence on whether the changes are brought about by the policy intervention per se or the changes are, in part or in whole, a natural result of the general growth and development of the country over the course of the duration of the study. This looks like an evaluation report unique to Brazil’s Health System and it is not possible to claim that all changes were brought about by government intervention without a good investigation of the counterfactuals, all other factors that may have contributed in the regionalization process.

This study reports expansion in health coverage and services and human resource expenditure but does not state any improvements, or lack thereof,  in the actual health status (Infant Mortality Rate, Maternal Mortality Ratio, Life Expectancy, etc.) of Brazil’s population as a result of these “innovative” health policy implementation. As I mentioned above, this information may be useful for local consumption but the study falls short of producing new knowledge that is reproducible, replicable, and applicable to improve public health policy or programmatic interventions beyond the study area.

Reviewer 2 Report

The purpose of this paper is to analyze the development of the public and private offer for the universalization of health services, specifically, for the progression of the public network in Brazil, for the period 2008 to 2015.

I list my comments as follows.

  • Please explain why you used the standard Z score of the original variables after applying the PCA.
  • Explain why you did not determine the z-score for the variables studied before PCA.
  • In Figure 3 mention the indicator used on the OY axis.
  • You mentioned that you used K-means clustering methodology. How did you determine the number of clusters in these 4 situations? In order to compute the dendrogram you have used the Hierarchical Cluster Analysis and not K-means.

Author Response

This manuscript is a resubmission of an earlier submission. The following is a list of the peer review reports and author responses from that submission.

Round 1

Reviewer 1 Report

I  commend the authors for their study title, “The Regionalization Process for Universal Health Coverage in Brazil (2008-2015)”

The authors have presented a report on the unique experience of almost a decade-long public health policy implementation in Brazil. This reviewer believes the report could be very important at the local, regional, and national levels for the public, private, and all other stakeholders in Brazil’s Health System.

This reviewer is not able to determine the design of the study,  the research question(s), the sampling method, the sampling unit, and the outcome variable (s,) and the independent variables that significantly influenced the outcome for this study.  The authors also do not provide a counterfactual evidence on whether the changes are brought about by the policy intervention per se or the changes are, in part or in whole, a natural result of the general growth and development of the country over the course of the duration of the study.

The study reports expansion in health covrage and services and human resource expenditure but does not state any improvements, or lack thereof,  in the actual health status (Infant Mortality Rate, Maternal Mortality Ratio, Life Expectancy, etc.) of the Brazil’s population as a result of these “innovative” health policy implementation.

Reviewer 2 Report

This analysis of the transition throught decentralization to regionalization of universal primary care in Brazil is fascinating. The most important result cited appears to be an increase in human resource capitalization at no additional cost to the public (as reported, unless I'm misinterpreting Table 5).  What societal entities are absorbing the additional costs?  This specific feature of the results should be stated or clarified by the authors.

There is too little in the introduction as the motivation for the restructing of healthcare administration in Brazil and its effect on overall population health.  What was/were the driving motivation(s)?

The missing information that is most relevant is the change in chronic illness  (asthma, allergic rhinitis, sinusitis, eczema, ear infections, otitis media, respiratory infections) and neurodevelopmental disorders (ASD, ADHD).   Behind the presented data is the onset of the Stork program in Brazil, which introduced mandatory widespread vaccination and pre-natal care across Brazil in 2011.   Microcephaly started thereafter, before Zika came to Brazil (per Mattos, WHO report).  Although originally attributed to Zika, but in the following years microcephaly has waned while Zika infections have become seasonal. The WHO report on microcephaly pre-dating the arrival of Zika to Brazil is important.  The de novo use of folic acid instead of methyl folate in a folic-acid naive population over this time period is likely to have had health consequences. Fetal death rates associated with vaccination during pregnancy would be important to know. Maternal deaths due to pregnancy and infant death rates (0-1 year, 2-3 years) would be important to map out. The report would be much more informative if some trends analysis of public health outcomes were included.

How healthcare is administered is not relevant in terms of costs to society unless the potential impact of those changes on overall human health is also considered.  It would serve the Brazilian public well to know of any health trends associated with the underlying model of distribution of health care services.

The report is written with exceptional clarity otherwise.

Relevant citations

https://www.nature.com/news/polopoly_fs/7.33594!/file/NS-724-2015_ECLAMC-ZIKA%20VIRUS_V-FINAL_012516.pdf

http://www.who.int/bulletin/online_first/16-170639.pdf   Here is some info on the prenatal Stork program:   http://www.brasildamudanca.com.br/en/saude/stork-network-provides-care-mothers-and-babies-during-pregnancy-through-birth   And reference to the mandate: http://www.thevaccinereaction.org/2016/02/tdap-vaccinations-for-all-pregnant-women-in-brazil-mandated-in-late-2014/

Reviewer 3 Report

The purpose of this paper is to analyze the development of the public and private offer for the universalization of health services, specifically, for the progression of the public network in Brazil, for the period 2008 to 2015.

I list my comments as follows.

  • According to the instruction for authors, the References must be numbered in order of appearance in the text (including table captions and figure legends) and listed individually at the end of the manuscript.
  • In the text, reference numbers should be placed in square brackets [ ], and placed before the punctuation; for example [1], [1–3] or [1,3]. For embedded citations in the text with pagination, use both parentheses and brackets to indicate the reference number and page numbers; for example [5] (p. 10). or [6] (pp. 101–105).
  • Why did you use the standard Z score of the original variables after applying the PCA? Why didn't you determine the z-score for the variables studied before PCA?
  • On page 7 you have “After selection of the variables, the standard Z score of the original variables was used to construct the dendrogram below”. I didn't find a dendrogram below. You refer to figure no. 1?
  • In Figure 2, mention the indicator used on the OY axis.
  • Tables 19, 20, 21 and 22 are not explained. You mentioned that you used K-means clustering methodology. How did you determine the number of clusters in these 4 situations?